



**The interaction between urbanization and aerosols during the haze event**
Miao Yu[1], Guiqian Tang[2], Yang Yang[1], Shiguang Miao[1], Yizhou Zhang[1], Qingchun Li[1]
1.  *Institute of Urban Meteorology, China Meteorological Administration, Beijing, China*
2.  *State Key Laboratory of Atmospheric Boundary Layer Physics and Atmospheric Chemistry (LAPC), Institute of*
7        *Atmospheric Physics, Chinese Academy of Sciences, Beijing 100029, China*

*Corresponding author*:
Guiqian Tang
*State Key Laboratory of Atmospheric Boundary Layer Physics and Atmospheric Chemistry (LAPC), Institute of*
*Atmospheric Physics, Chinese Academy of Sciences, Beijing 100029, China*



## Abstract

The interaction between aerosols and urbanization during the haze event was investigated using the Rapid-
Refresh Multiscale Analysis and Prediction System-Short Term (RMAPS-ST). The mechanisms of the impacts
of aerosols and urbanization were also analyzed and quantified. Aerosols reduce urban-related warming during
the daytime, and the warming decreased by 30 to 50% as the concentration of $PM_{2.5}$ increased from 200 to
400 $\mu g \cdot m^{-3}$. Aerosols enhance the urban-related warming at dawn, with an increase of approximately 28%,
which is important for haze formation. Urbanization reduced the aerosol-related cooling effect by
approximately 54% during the haze event, and the strength of the impact changed little with increasing aerosol
content. The impact of aerosols on urban-related warming is more significant than the impact of urbanization
on aerosol-related cooling. Aerosols decreased the urban-impact on the mixing layer height by 148% and on
the sensible heat flux by 156%. Furthermore, the aerosols decreased the latent heat flux, and the impact was
reduced by 48.8% by urbanization. The impact of urbanization on the transport of pollutants is more important
than that of aerosols. The interaction between urbanization and aerosols may enhance the accumulation of
pollution and weigh against diffusion.

## 1 Introduction


In recent years, heavy haze pollution events have occurred more frequently in densely populated urban areas,
such as the Beijing-Tianjin-Hebei region (BTH region) and Yangtze River Delta region of China, which has
caused increasingly serious adverse effects on transportation, the ecological environment and human health
(Zhao et al., 2012; Wu et al., 2010; Liu et al., 2012). A statistical analysis of the variation in haze days in
Beijing over the past 10 years shows that the number of haze days has significantly increased (Chen and Wang,
2015; Zhai et al., 2019). The average annual number of haze days was 162 in 1981-1990, 167 in 1991-2000,
and 188 in 2001-2010. The conditions for the formation of heavy haze weather in the BTH region are very
complex (Miao et al., 2017; Wei et al., 2018; Ren et al., 2019). Atmospheric pollutant emissions,
meteorological conditions, terrain, and urban high-density human activities are all important conditions for
the formation of heavy haze weather (Zhu et al., 2018). However, meteorological conditions are becoming the
most critical conditions for the development of heavy haze pollution weather when there is little change in
atmospheric pollutant emissions (Wang et al., 2020; Pei et al., 2020).

The characteristics of the atmospheric boundary layer structure determine the horizontal fluidity, vertical





diffusion ability, stability and capacity (mixed layer thickness) of the atmosphere, which are the main factors
affecting the formation, intensity and duration of haze and atmospheric pollution (Guo et al., 2016). Coulter
R L. (1979) indicated that the height of the mixing layer would affect the concentration and diffusion of
pollutants, which has been one of the most important physical parameters in atmospheric numerical models
and atmospheric environment evaluations, and urbanization and aerosols have been proven to influence the
boundary layer height (Tao et al., 2015).

Urbanization, as the most drastic means by which human activities transform the environment, has had an
important impact on regional climate and weather processes (Miao et al., 2011; Yu and Liu, 2015; Yu et al.,
2017). Existing research suggests that there are three main ways by which urbanization influences the climate
(Oke, 1982 and 1995). The change with land use from natural surfaces to impervious underlying surfaces in
association with urbanization alters the surface albedo and roughness, which results in the formation of urban
heat islands (UHIs) (Taha, 1997; Folberth et al., 2014). This leads to a change in the surface energy balance
and the form of the thermal difference between urban and rural areas and further changes the boundary layer
structure (Grimmond, 2007; Li and Bou-Zeid, 2013). Second, thermal differences further lead to heat island
circulation, which can influence the local circulation of synoptics and the transport of pollutants (Crutzen,
2004). Anthropogenic aerosols and heat from the development of transportation and industry are also
important parts of urban impacts on climate. However, aerosols can reduce the decrease in shortwave radiation
and cause cooling at the surface and enhance static stability, which is opposite to the effects of urbanization
(Grimmond, 2007; Cruten, 2004). Furthermore, aerosols may increase longwave radiation in urban areas
because they are likely to absorb and emit more energy than water vapor or greenhouse gases under certain
conditions (Jacobson,1998; Rudich et al., 2007). There have been few studies on the mechanism of the
interaction between urbanization and aerosols, although many studies focus on their respective effects.
Accordingly, the interaction between urbanization and aerosols is important for studying regional climate.

Researchers are increasingly aware of the importance of the interaction between urbanization and aerosols. A
very important study by Cao et al. 2016 was the first attempt to determine the effects of aerosols on
urbanization and indicated that aerosols can increase the nighttime UHI effect using a climate model. Yang et
al. 2020 obtained different results when using observational data to perform similar research in the BTH region.

More detailed research needs to be performed by combining observational data and modeling because the





conclusions may vary depending on the scale (Xu et al., 2019). Other illuminating work with regional models
showed that the combined effect of UHIs and aerosols on precipitation depends on synoptic conditions (Zhong
et al., 2015). However, for winter haze, Zhong et al. (2017) evaluated the urban impact on air quality and
indicated that urbanization can increase ventilation in daytime and increase aerosol emissions, which
outweighs the UHI effect.

However, very few studies have quantified the individual effects of urbanization-induced UHIs and aerosols
with elevated emissions on the formation and development of haze in metropolitan areas. A difficulty is that
the radiative forcing of aerosols is not a prognostic variable in most climate models (Cao et al. 2016). Some
regional models such as WRF-Chem can overcome this problem by parameterizing aerosols to aerosol optical
depth (AOD) in some specific radiation schemes. Tao et al. 2015 and Zhong et al. 2018 have made some
progress in this area, and their results also indicate that the regional model can be used as an effective way to
study the interaction between urbanization and aerosols. However, a quantitative evaluation of urban impacts
on aerosols and aerosol impacts on urban-impact at the same time in metropolitan areas has not been attempted.

In this study, the Rapid-Refresh Multiscale Analysis and Prediction System-Short Term (RMAPS-ST) was
used to investigate the mechanism of the influence of the above two factors in a typical winter haze event. The
objective of this study is 1) to quantify impact of urban on aerosols and impact of aerosols on urbanization
respectively and 2) to obtain a better understanding of the interaction between urbanization and aerosols and
its influence mechanism on the boundary layer structure and haze transmission during the typical winter haze
events in the BTH region. This research will help to improve air quality under the continuous
urbanization and sustainable development of large cities.

**2 Methods**
**2.1 Observational data**
Four kinds of observational data were used in this study to reveal the synoptic situation of haze events and
perform model evaluation. Meteorological data from 309 national basic weather stations in the BTH region
were provided by the China Meteorological Administration (http://data.cma.cn/). The locations of the national
basic weather stations are shown in Fig 1 (red dots). The mass concentrations of fine particulate matter ($PM_{2.5}$)
were recorded by 251 environmental monitor stations managed by the Ministry of Ecology and Environment
of the People's Republic of China (http://hbk.cei.cn/aspx/default.aspx) (Fig 1, black dots). Radiation and


surface heat flux data were obtained from the Beijing meteorological tower (39.97°N, 116.37°E), which is
325 m high and operated by the Institute of Atmospheric Physics (IAP), Chinese Academy of Sciences (CAS).
The heat flux data were measured by a fast response eddy covariance sensor system that was sampled at 10
Hz using CR500 (Campbell Scientific Inc., USA). The radiation data were provided by Kipp & Zonen
(Netherlands) four-component unventilated CNR1 radiometers. Radiation and surface flux data from 140 m
of the tower were used in this study. The mixing layer height (MLH) and backscattering coefficient were
measured by enhanced single-lens ceilometers (Vaisala, CL51, Finland) deployed by the IAP. Backscattering
coefficient profiles were calculated by reference to the attenuation strobe laser LiDAR technique (910 nm),
which is cited in Tang et al. (2015).

**2.2 Model description and experimental design**
The model used in this study is the latest available version of RMAPS-ST, developed by the Institute of Urban
Meteorology, China Meteorological Administration. RMAPS-ST is based on the Weather Research and
Forecasting (WRF v3.8.1) model (Skamarock et al., 2008) and its data assimilation system (WRFDA v3.8).
The simulation domain was centered at 37.0 °N, 105.0 °E and implemented with two nested grids with
resolutions of 9 and 3 km for two domains (D1 and D2, respectively) (Fig 1a). The model performance was
verified and RMAPS-ST runs operationally (Fan et al., 2018). The assimilation began every three hours, and
the assimilated data included automatic meteorological station data, sounding data and radar data when
available. The model settings are shown in Table 1. The simulation started at 0000 LST and ran from 15 to 23
December 2016 with hourly output.

The urban impact was represented by a high-resolution (30 m) land use map interpreted from Landsat
Thematic Mapper satellite data for 2015 in Beijing. The urban canopy parameters were optimized according
to Miao and Chen (2014). The impact of aerosols was represented by adding the hourly distribution of AOD
in the RRTMG radiation scheme. The AOD was extracted from the output of RMAPS-Chem (Zhao et al.,
2019; Zhang et al., 2018) for the BTH region, which is shown in Fig 1b. Anthropogenic emission data were
obtained according to the Multiresolution Emission Inventory for China (2012) (http://www.meicmodel.org/)
with a resolution of 0.1°×0.1°. The simulated distribution of AOD in Beijing has been verified to be
satisfactory when compared to the observed vertical profile of the backscattering coefficient (Fig 2a and b).
The correlation of AOD and the column backscatter coefficient is 0.76 (Fig 2c). Four tests were designed to
investigate the impacts of aerosols and urbanization on typical haze events. Test 1: Both urban and aerosol





impacts were considered in the simulation. We updated the grid AOD distribution hourly as the input field for
the RRTMG radiation scheme in Domain 2. Test 2: Only aerosol impact was considered in the simulation,
and we replaced the urban grid with cropland to shield the impact of urbanization. Test 3: Only urban impact
was considered, and the direct radiative forcing of aerosols was not considered in the simulation. Test 4: Both
urban and aerosol impacts were not considered in the simulation.

The model evaluation results for the four tests are shown in Table 2. As the service operational system, the
RMAPS-ST model assessment report indicated that the model performance was satisfactory (Fan et al. 2018).
We evaluated not only the conventional meteorological variables (including temperature, humidity and wind
speed) but also unconventional but important variables for this study (including radiation and surface heat
flux). A total of 309 meteorological station data points were used to evaluate the conventional variables. The
unconventional variables were evaluated according to the observational data from 140 m of the Beijing
meteorological tower. Test 1 was found to be the best simulation and considers both the urban and aerosol
impacts.

## 3    Results

### 3.1 Weather analysis

A typical continuous severe heavy haze occurred from the 15th to 22nd of December 2016 in the BTH region.
Three stages dominated by three different synoptic patterns controlled the formation of this haze. In the first
stage, northwest airflow in front of a ridge of high pressure was observed in the BTH region at a height of 700
to 500 hPa and in eastern China at a height of 850 hPa on the 15th to 16th of December, which induced a sharp
warming pattern (Fig 3a and b). At the surface, Beijing was located under the front of the high pressure system
to under the southwest airflow in front of the low pressure system (Fig 4), which favored pollutant transport
from Hebei Province to Beijing. From the 17th to the night of the 18th, the control system turned to the latitude
circulation at 700 to 500 hPa over the BTH region (there was a trough line south of 40°N at 2000 LST on the
17th and 18th) (Fig 3c). There was a northwest wind located north of 40°N and a southwest wind located south
of 40°N at 850 hPa (Fig 3d). The near surface was controlled by the northeast airflow located in the inverted
trough of the low pressure. The weak convergence of the high trough cooperates with the low pressure at the
surface, leading to continuous pollution accumulation near the surface. Under this weather situation, the near-
surface temperature began to continuously increase from the 16th to 18th, and the specific humidity also
correspondingly increased (Fig 5a). The near-surface wind speed and pressure decreased during this period





(Fig 5b). The concentration of $PM_{2.5}$ gradually increased from the $16^{th}$, and the average concentration of $PM_{2.5}$
reached 200 $\mu g \cdot m^{-3}$ on the $18^{th}$. The density of ozone obviously decreased from the $16^{th}$ (Fig 5c).

The MLH significantly declined from the $16^{th}$, and the diurnal circle almost disappeared during this period,
accompanied by a visibility reduction but diurnal variation (Fig 5d). The downward shortwave radiation and
the net radiation gradually decreased from the $16^{th}$ to the $18^{th}$, which directly influenced the variation trend of
ozone (the maximum density of ozone was less than 110 $mg \cdot m^{-3}$), while there was little change detected in
longwave radiation (Fig 5e). The observed sensible heat flux also decreased from the $16^{th}$ to the $19^{th}$ although
the temperature increased, which means that the heat exchange became weaker in the vertical direction, while
the latent heat flux changed little (Fig 5f). Southwest airflow was again captured by a wind profiler on the
night of the $18^{th}$ and the transport layer occurred from 300 to 1500 m, which differs from the previous surface
transport pattern (Fig 4).

In the second stage, the important change occurred in the morning of the $19^{th}$ of December, when the control
system turned to the northwest airflow on the front of the trough over the BJH region at 500 to 850 hPa (Fig
3e and f). After 2000 LST on the $19^{th}$, obvious warming occurred again at 850 hPa in eastern China (Fig 3h).
However, the near-surface maximum temperature and diurnal range in Beijing significantly decreased but with
high specific humidity during the $20^{th}$ to the $21^{st}$ (Fig 5a). According to the surface weather map, the control
system turned to the southwest at 1400 LST on the $19^{th}$, and a large-scale southeast wind appeared in eastern
Beijing after 2000 LST, which induced wide advection fog formation during the night (Fig 3g). Due to the
influence of the southwest airflow on the tough at 500 hPa, the inverted trough moved east, and Beijing was
located in the southeast wind zone. The near-surface pressure increased slightly, and the wind speed remained
low at approximately 1 $m \cdot s^{-1}$ (Fig 5b). The synoptic system caused the $PM_{2.5}$ concentration to peak
(approximately 400 $\mu g \cdot m^{-3}$ on average and above 500 $\mu g \cdot m^{-3}$ observed at some stations) and was maintained
from the $20^{th}$ to the $21^{st}$ in the BTH region. The visibility was less than 400 meters, and the diurnal circle
disappeared (Fig 5d). The decrease in the downward shortwave and net radiation was more pronounced than
that in the previous three days (Fig 5e). The sensible heat flux also decreased, and the diurnal circle almost
disappeared from the $19^{th}$ to the $20^{th}$ (Fig 5e).

It was not until the strong cold air moved southward in the early morning of the $22^{nd}$ when the whole
atmosphere converted to the northwest stream. The air pollutants were completely removed in the third stage.



### 3.2 Interaction between the impacts of urbanization and aerosols on haze events

Four impacts were analyzed as following. Urban impact under the aero scenario (UI_aero) was represented
by the results of Test 1 minus those of Test 2; urban impact under the no-aero scenario (UI_noaero) was
represented by the results of Test 3 minus those of Test 4; The impact of the urbanization scenario was
represented by the results of Test 1 minus those of Test 3 (AI_urban); the impact without urbanization was
represented by the results of Test 2 minus those of Test 4 (AI_nourban). The interaction between urbanization
and aerosols on local meteorological and regional transportation was discussed.

### 3.2.1    The impact on the local area

Temperature is one of the most sensitive variables affected by urbanization and aerosols and is also the most
concerning variable. The impact of urbanization on the near-surface temperature in the Beijing area displays
diurnal variation features. The warming induced by urbanization was dominant at night. The urban impact
was obviously decreased under the aerosol scenario by comparing the results of UI_aero and UI_noaero,
especially in the daytime (Fig 6a, red lines). The urban impact always showed a positive contribution to the
temperature during the whole day under the no-aerosol scenario, while the urban impact became slightly
negative with the aerosol scenario in the daytime. The maximum difference between UI_aero and UI_noaero
occurred on the $20^{th}$ and $21^{st}$, when the AOD value reached its maximum, and the difference almost
disappeared on the $15^{th}$ and $22^{nd}$, with a small AOD (Fig 2b). The results indicate that the impact of
urbanization on temperature is reduced by aerosols, which is consistent with the findings of Yang et al. 2020.
The average urban impact on temperature in Beijing during the $16^{th}$ to $19^{th}$ with a $PM_{2.5}$ concentration of
approximately 200 $mg·m^{-3}$ was a reduction of 0.42°C according to UI_aero and of 0.60°C according to
UI_noaero. This means that aerosols reduce the urban impact on temperature by 30%. When the concentration
of $PM_{2.5}$ reached 500 $mg·m^{-3}$ from the $20^{th}$ to the $21^{st}$, the aerosols reduced urbanization-related warming by

236 53.5%.


The impact of aerosols on temperature is negative and without a diurnal circle under the urbanization scenario
for the whole day (Fig 6a, blue lines). However, the impact of aerosols captured by AI_nourban is more
significant and displays a diurnal circle. Another important observation is that the impact of aerosols on
temperature under the no-urban scenario is not always negative. There is a slight warming period at dawn in
the AI_nourban scenario, which maybe because the longwave radiation is increased (Jacobson,1998; Rudich





et al., 2007). The average impact of aerosols on temperature in Beijing was -0.16°C with urbanization and -
0.34°C without urbanization from the 16th to the 19th. The impact of aerosols was -0.19°C with urbanization
and -0.43°C from the 20th to the 21st. Urbanization decreased the impact of aerosols by 53% under moderate
pollution and by up to 56% under heavy pollution. Two different impacts of aerosols on urban-related warming
were observed. There was a reducing effect in the daytime with a strength of approximately 30 to 50% of the
concentration and an increasing effect occurred at dawn with a strength of approximately 28%. Urbanization
reduced the aerosol-related cooling effect by approximately 54%.

The observed specific humidity continued to increase as the aerosol concentration increased (Fig 5b) and is
closely related to the UHI effect and aerosol composition (Zhang et al. 2010; Sun et al., 2013; Wang et al.,
2020). The specific humidity also increased with urbanization throughout the day (Fig 6b, red lines). Similar
to temperature, urbanization had a more pronounced impact on specific humidity at night. The average urban
impact on specific humidity was 0.0366 g·kg$^{-1}$ according to UI_aero and 0.0478 g·kg$^{-1}$ according to UI_noaero
during the 16th to 19th and 0.0308 and 0.0448 g·kg$^{-1}$ during the 20th to 21st. Aerosols not only reduced the urban
impact on the average daily specific humidity by 23.43% but also reduced the diurnal range of specific
humidity.

In contrast to urbanization, aerosols were found to reduce the specific humidity (Fig 6b, blue lines). The impact
of aerosols under the urbanization scenario was small and without a diurnal pattern. However, their impact
under the no-urban scenario was more distinct and with a diurnal circle. The average impact of aerosols on
specific humidity was -0.0088 g·kg$^{-1}$ according to AI_urban and -0.0136 g·kg$^{-1}$ according to AI_nourban
during the whole study period. Urbanization reduced the impact of aerosols on specific humidity by 35.3%.
The impacts of urbanization and aerosols on humidity were slightly greater than those of aerosols on urban
impacts.

There was no effect of urbanization on downward shortwave radiation according to both UI_aero and
UI_noaero (Fig 6c, red lines), although the value is not absolutely related to aerosols because of model
uncertainty. Aerosols reduce the downward shortwave radiation in the daytime, and there are few differences
between AI_urban and AI_nourban.

The average decrease in shortwave radiation caused by aerosols was approximately 7% of the total downward





shortwave radiation during the 16th to the 20th and up to 17% when the $PM_{2.5}$ was greater than 400 µg·m$^{-3}$.
The urban impact increased the longwave radiation in the nighttime according to UI_aero, while the impact
of urbanization was always positive for longwave radiation during the study period according to UI_noaero
(Fig 6d, red lines). Because it is closely related to temperature, the urban impact on long wave radiation was
also reduced by aerosols, with reductions of 83.3% from the 16th to the 19th and of 96.6% from the 20th to the
21st. The impact of aerosols on longwave radiation is smaller than that of shortwave radiation, and there was
a slight decrease captured by AI_urban with an increase from noon on the 20th to nighttime on the 21st. The
impact of aerosols decreased the longwave radiation captured by AI_nourban during the 16th to the 20th and
increased it on the night of 21st (Fig 6d, blue lines). Urbanization reduced the impact of aerosols on longwave
radiation by 66.9% while aerosols reduced the urban impact on longwave radiation by 89.2%. The impacts of
urbanization and aerosols on longwave radiation are unimportant because they are both smaller than 2 W·m$^{-}$
$^{2}$.

The change in radiation further alters the MLH. Previous studies suggest that MLH is important for the
diffusion of pollutants and haze formation (Sun et al. 2013; Quan et al. 2014). Previous studies on urbanization
indicated that urban-induced warming will increase the MLH during the daytime (Wang et al., 2007; Miao et
al. 2012), and the results of UI_noaero show the same pattern. However, when we introduced aerosols into
the simulation, urbanization was found to decrease the MLH in the daytime according to UI_aero. The impact
of aerosols decreased the average urbanization by 148% during the haze event (Fig 6e, red lines). Aerosols
significantly decreased the MLH in daytime according to both AI_urban and AI_nourban (Fig 6e, blue lines).
Urbanization decreased the impact of aerosols on MLH by 57.84% during the haze event.

Urban land use change directly alters the surface heat flux. Urbanization increased the sensible heat flux
according to UI_noaero but decreased the sensible heat flux according to UI_aero (Fig 6f, red lines). The
impact of aerosols in reducing the urban impact on sensible heat flux was 156% during the haze event.
Aerosols reduced the sensible heat flux according to both AI_urban and AI_nourban (Fig 6f, blue lines). The
maximum impact of aerosols was on the 21st, with the maximum AOD. The impact of urbanization reduced
the impact of aerosols on sensible heat flux by 59.3%.

There was little effect of urbanization on latent heat flux because the observed latent heat flux in urban areas
was small (Fig 6g, red lines, and Fig 5e). Aerosols decreased the latent heat flux, and the impact increased



with increasing AOD (Fig 6g, blue lines). The impact of urbanization reduced the impact of aerosols on the
latent heat flux by 48.8%.

In general, the impact of aerosols on urban impacts is more important than the impact of urban impacts on
aerosol impacts in terms of local effects.

**3.2.2 Effects on regional circulation**
There are few valuable findings from the diurnal average wind speed analysis because the average wind speed
was low during the haze event. Wind speed is likely to become more meaningful in the spatial analysis of
wind vectors. There are two main transmission processes of pollution from Hebei Province to Beijing in this
haze process according to the weather map and wind profile analysis (Fig 4). Accordingly, the diurnal pattern
of $PM_{2.5}$ in Beijing (Fig 5c) also displays two increasing processes on the 16th and 19th (from 1800 to 2400
LST). The observed near-surface wind vector displays these two pollutant transport processes (Fig 7). In the
first processes, obvious aerosol transport began on the night of the 15th and continued to the night of the 16th
(Fig 6). The southwest wind dominated most of the southern part of Hebei Province. The transmission flux
was strong in the daytime on the 16th, leading to the concentration of $PM_{2.5}$ continuing to increase in Beijing
and in its transmission path. The wind speed remained low from the 17th to the 18th in most of the plain area,
and the concentration of $PM_{2.5}$ continued to increase in the southwest and northeast of Hebei Province. The
second processes began at 1400 LST on the 19th and the south wind dominated the south of Beijing and turned
to the southwest in Beijing at 1400 to 1800 LST. The dominant wind direction turned to the southwest at 2200
LST in the southern part of Hebei Province with a rapid increase in the concentration of $PM_{2.5}$.

Most industrial aerosols in Beijing are transported from the southwest and northeast of Hebei Province due to
the control of pollutant discharge in the Beijing area during haze events. Therefore, the impact of urban areas
and aerosols on transport, namely wind fields is very important for air quality in Beijing. The modeling results
show that urbanization not only increased the temperature in urban areas (Fig 8a and b) but also increased the
average south-wind transport flux in the two main transmission processes of pollution in the southwest area
of Beijing (Fig 8a and b). The transmission flux captured by UI_noaero is stronger than that captured by
UI_aero. The local cyclonic circulation induced by urbanization further induces upward movement, which is
beneficial to diffusion conditions. Although aerosols decrease the transmission flux induced by urbanization,
the strength of local cyclonic circulation is also reduced by aerosols. Furthermore, the aerosols reduced the



temperature in most of the plain area in Hebei Province (Fig 8c and d). Urbanization decreases the impact of aerosols on temperature. There was no local or systemic effect on the wind field captured by either AI_urban or AI_nourban.

Taylor diagrams were used to analyze the relative contributions of urbanization and aerosols over time (Fig 9). The daily mean difference in these four types of impact (UI_aero, UI_noaero, AI_urban, and AI_nourban) over the eight days in the Beijing area is shown by Taylor diagrams. UI_noaero shows that temperature continues increasing from Day 1 to Day 5 and reaches a maximum on Day 7. The variation in temperature according to UI_urban is smaller. This means that the effect of urbanization on temperature is decreased by aerosols. Temperature increases from Day 1 to Day 7 according to AI_urban, while AI_nourban shows an increase from Day 3 to Day 7. The reduction of the urban impact on temperature by aerosols was more important than the reduction of aerosol impact on temperature by urbanization (Fig 9a). The effect of aerosols on urban impacts on temperature was more important than urban impacts on the effects of aerosols on temperature (Fig 9a). Specific humidity continued increasing from Day 1 to Day 5 according to UI_noaero, while the variation in specific humidity was small according to UI_aero (Fig 9b). Similar to what was observed for temperature, reducing the urban impact on specific humidity by aerosols is more important than reducing aerosol impacts by urban areas. The ventilation coefficient (VC) in UI_aero showed little change over these eight days, and this coefficient showed increases on Days 2, 3, 5, and 6 and decreases on Days 4, 7, and 8 according to UI_noaero. The reduction of the urban impact on the VC by aerosols is more important than the reduction of the impact of aerosols by urbanization. The analysis of shortwave radiation also provided the same conclusion that the reduction in the urban impact on the daily mean by aerosols was more important than the reduction of the impact of aerosols by urbanization (Fig 9d).

### 3.2.3 Impacts on the vertical distribution

In the period from 0000 LST to 0800 LST on the 16th to 20th, there was an interesting phenomenon that temperature was a slightly larger in UI_aero than in UI_noaero, and the urban impact reached a maximum at the same time. Such an outcome is easy to overlook if the analysis only focuses on the daily average. Therefore, a detailed vertical temperature and wind field analysis of the four addressed scenarios (UI_aero, UI_noaero, AI_urban, and AI_nourban) was used to determine the mechanism behind this finding (Fig 10).

The impact on warming by urbanization reached 350 m in UI_aero and 450 m in UI_noaero (Fig 10a and b).





Aerosols not only increased the warming impact induced by urbanization but also reduced the warming height.
Aerosols increase the near-surface warming effect induced by urbanization because of the absorption of
longwave radiation. Although absorption by aerosols was always observed during the study period, the impact
increased with the increase in longwave radiation induced by urbanization. Therefore, the warming effect of
aerosols may dominate at night in the near-surface layer. This further induces the urban-related warming to
increase and compress this effect to a lower height with a lower MLH in UI_aero than in UI_noaero (Fig 10a).
The aerosols reduced the temperature below 450 m in the urban area of Beijing (Fig 10c and d) and the cooling
effect was reduced by urbanization below 450 m. Urbanization also reduces the near-surface west wind
induced by aerosols in urban areas because of the drag caused by buildings.

**4 Conclusion**
A typical persistent haze process occurred on the 15th to 22nd of December 2016 in the BTH region. The
average concentration of $PM_{2.5}$ was approximately 200 $\mu g \cdot m^{-3}$ and the maximum was greater than 400 $\mu g \cdot m^{-}$
$^3$. The interaction between aerosols and urbanization on haze events were investigated in this study. Four tests
were designed using RMAPS-ST to study the mechanism of the impacts of aerosols and urbanization
respectively.

Two different impacts of aerosols on urban-related warming were found. A reducing effect occurred during
the daytime, and the strength was approximately 30 to 50% of the concentration. An increasing effect occurred
at dawn, and the strength was approximately 28%, which is important for haze formation. The combined effect
was a reducing effect on the daily mean of urban-related warming. Urbanization reduced the aerosol-related
cooling effect by approximately 54% during the haze event, and the strength of the impact changed little with
increasing aerosol content. The impact of urbanization on the effect of aerosols on humidity is slightly larger
than the impact of aerosols on urban impact. Aerosols reduce the average downward shortwave radiation from
7% to 17% with concentrations of $PM_{2.5}$ of 200 to 400 $\mu g \cdot m^{-3}$. There is no urban impact on downward
shortwave radiation or the impact of aerosols on shortwave radiation. The impacts of urban areas and aerosols
on longwave radiation are both smaller than 2 $W \cdot m^{-2}$. A more significant impact of aerosols is on the MLH
and sensible heat flux. The decrease in urban impact caused by aerosols reaches 148% for MLH and 156%
for sensible heat flux. These values are much larger than those for urbanization, which reduces the impact of
aerosols on the MLH and sensible heat flux. There is little urban impact on latent heat flux. However, aerosols
decreased the latent heat flux, and the impact was reduced by 48.8% by urbanization. In general, the impact


of aerosols on urban impact is more important than the impact of urbanization on aerosol impacts in terms of
regional averages.

Urbanization increased the wind speed southwest of the Beijing area and the local cyclonic circulation in the
urban area of Beijing during the two main transmission processes. Although aerosols reduced the urban-related
southwest transmission, they made the diffusion conditions worse in urban areas. The impact of urbanization
on wind fields, namely, the transport of pollutants, is more important than that of aerosols. However, the
interaction between urbanization and aerosols may enhance the accumulation of pollution and weigh against
diffusion.

The impact of aerosols on urban-related warming is more significant than the impact of urbanization on
aerosol-related cooling according to spatial statistical analysis. Similar results were found for absolute
humidity, the VC and shortwave radiation. Aerosol-related warming is dominant at dawn in the near-surface
layer. Aerosols increase urban-related warming and reduce the impact height of urban-related warming. This
further enhances stability and reduces the MLH.

In this study, it was easier to distinguish the impacts of aerosols and urbanization by using the RMAPS-ST
with AOD hourly input than with RMAPS-Chem to investigate the impact of aerosols. One reason for this is
that the model performance of RMAPS-ST is much better than that of RMAPS-Chem in meteorological fields.
Although real-time feedback in modeling is not provided, RMAPS-ST is more efficient and more suitable for
short-term operational forecasting.

**Data availability**
The data in this study are available from the corresponding author upon request (tgq@dq.cern.ac.cn).

**Author contribution**
Miao Yu designed the research and wrote the paper. Guiqian Tang conducted the measurements and reviewed
the paper. Yang Yang conducted modelling tests. Qingchun Li did synoptic analysis. Shiguang Miao and
Yizhou Zhang reviewed and commented on the paper.




**Competing interests**

The authors declare that they have no conflicts of interest to disclose.

Table 1 RMAPS-ST model settings.

| WRF v3.8.1 | D01 | D02 |
|---|---|---|
| Horizontal grid | 649×400 | 550×424 |
| Grid horizontal spacing (km) | 9 | 3 |
| Vertical layers | 49 | |
| PBL | YSU (Hong et al., 2006) | |
| Microphysics | Thompson (Thompson et al., 2008) | |
| Cumulus | Kain-Fritsch (Kain, 2004) | None |
| LW Radiation | RRTMG | |
| SW Radiation | RRTMG | |
| LSM | Noah LSM+SLUCM | |
| Urban parameter values | Modified according to Miao and Chen (2014) | |

Table 2 Model evaluation (RMSE and BIAS) for the four tests.

| | Test 1 | | Test 2 | | Test 3 | | Test 4 | |
|---|---|---|---|---|---|---|---|---|
| | RMSE | BIAS | RMSE | BIAS | RMSE | BIAS | RMSE | BIAS |
| **Temperature** | **1.27** | **0.35** | 1.45 | -0.73 | 2.12 | 1.04 | 1.78 | -0.45 |
| **Specific humidity** | **0.26** | **-0.015** | 0.31 | 0.019 | 0.34 | -0.05 | 0.29 | 0.03 |
| **Wind speed** | **1.62** | **0.97** | 2.08 | 1.68 | 1.85 | 1.04 | 1.96 | 1.67 |
| **Shortwave** | **40.91** | **11.85** | 40.95 | 11.89 | 47.35 | 17.45 | 46.26 | 16.45 |
| **Longwave** | **51.39** | **-43.65** | 51.32 | -44.45 | 51.24 | **-43.53** | 52.76 | 44.97 |
| **Sensible heat flux** | **8.09** | **-1.19** | 9.13 | -3.92 | 9.34 | -3.43 | 12.3 | -6.17 |
| **Latent heat flux** | **14.09** | **-5.75** | 14.52 | -5.95 | 14.85 | -5.87 | 16.76 | -6.23 |





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





**Figure**

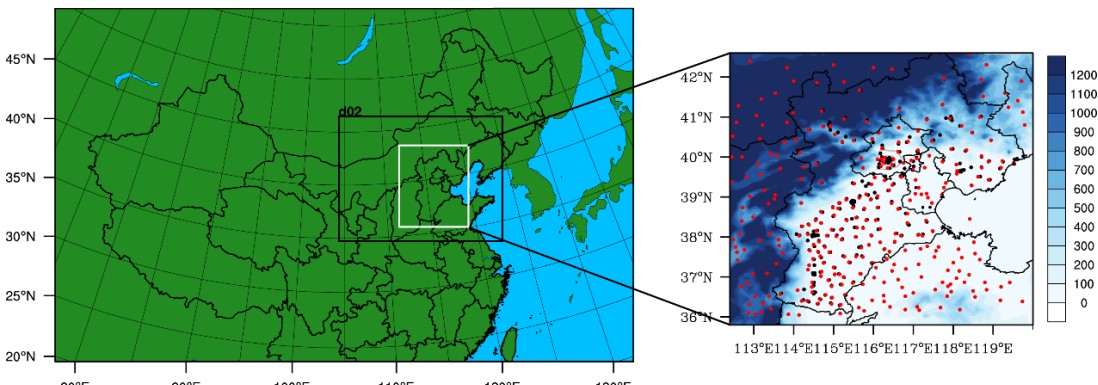


Figure 1 Domain configuration of RMAPS-ST and the location of the study area, indicated by the white solid line. The black
dots indicate the locations of the 251 environmental monitoring stations, and the red dots represent the 309 meteorological
stations in the Beijing-Tianjin-Hebei region, where the gray loop lines show the locations of the second to sixth ring roads.


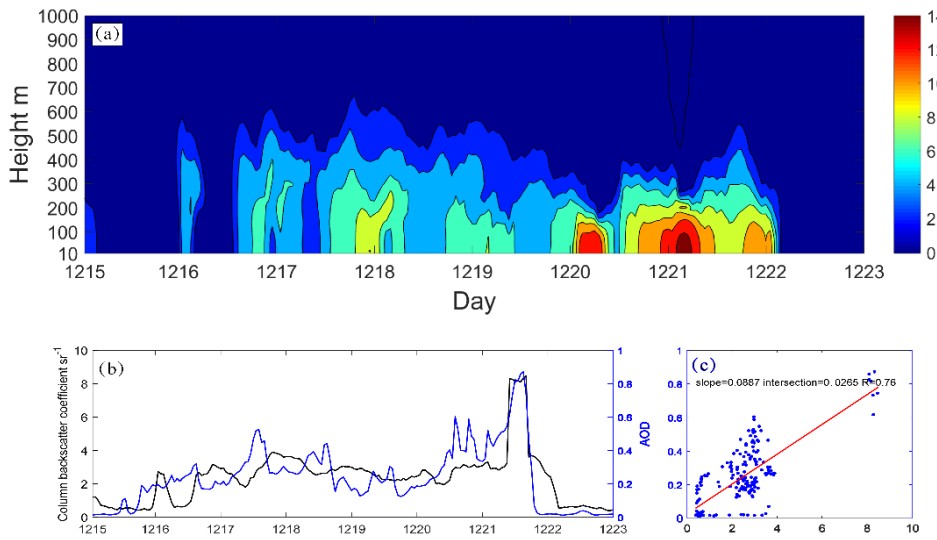

Figure 2 (a) Hourly backscattering coefficient (shading; mm·sr$^{-1}$) observed by single-lens ceilometers (39.97°N, 116.37°E)
from the 15$^{th}$ to 23$^{rd}$ of December; (b) hourly column backscatter coefficient (black line; sr$^{-1}$) and AOD used in modeling for
Beijing (blue line) and (c) their correlations.





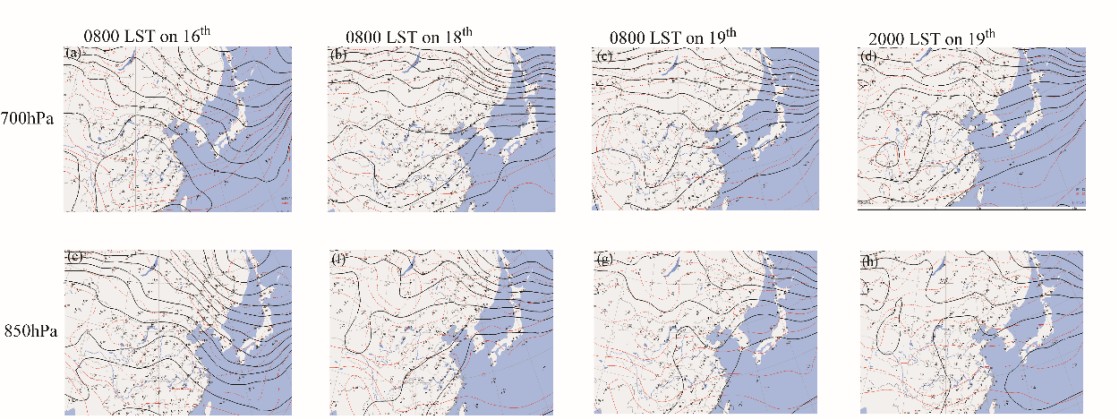

Figure 3 Weather maps. (a) 0800 LST on the 16th at 700 hPa; (b) 0800 LST on the 18th at 700 hPa; (c) 0800 LST on the 19th at 700 hPa; (d) 2000 LST on the 19th at 700 hPa; (e) 0800 LST on the 16th at 850 hPa; (f) 800 LST on the 18th at 850 hPa; (g) 0800 LST on the 19th at 850 hPa; (h) 2000 LST on the 19th at 850 hPa.

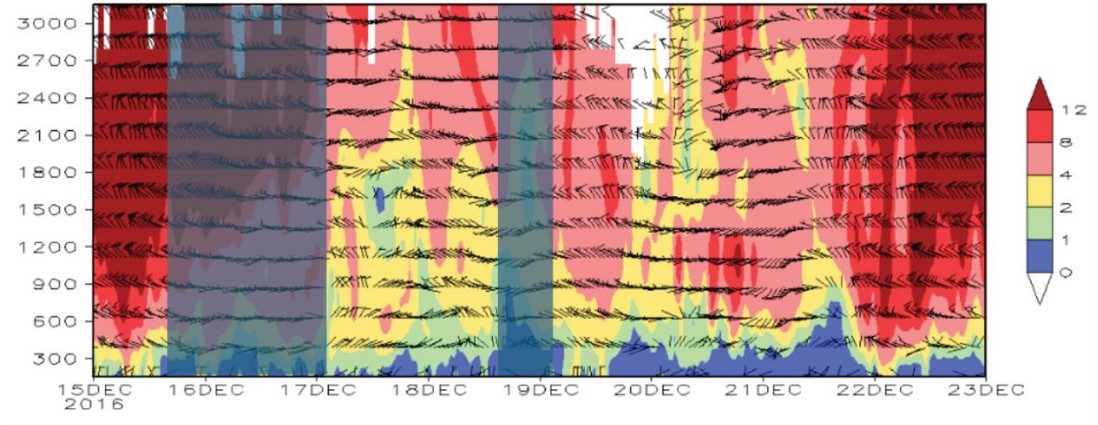

Figure 4 Hourly wind profile from the 15th to 23rd of December. Wind speed (shading; m·s⁻¹) and horizontal wind field (vector; m·s⁻¹). The shaded parts show the two periods of south wind conveyance.

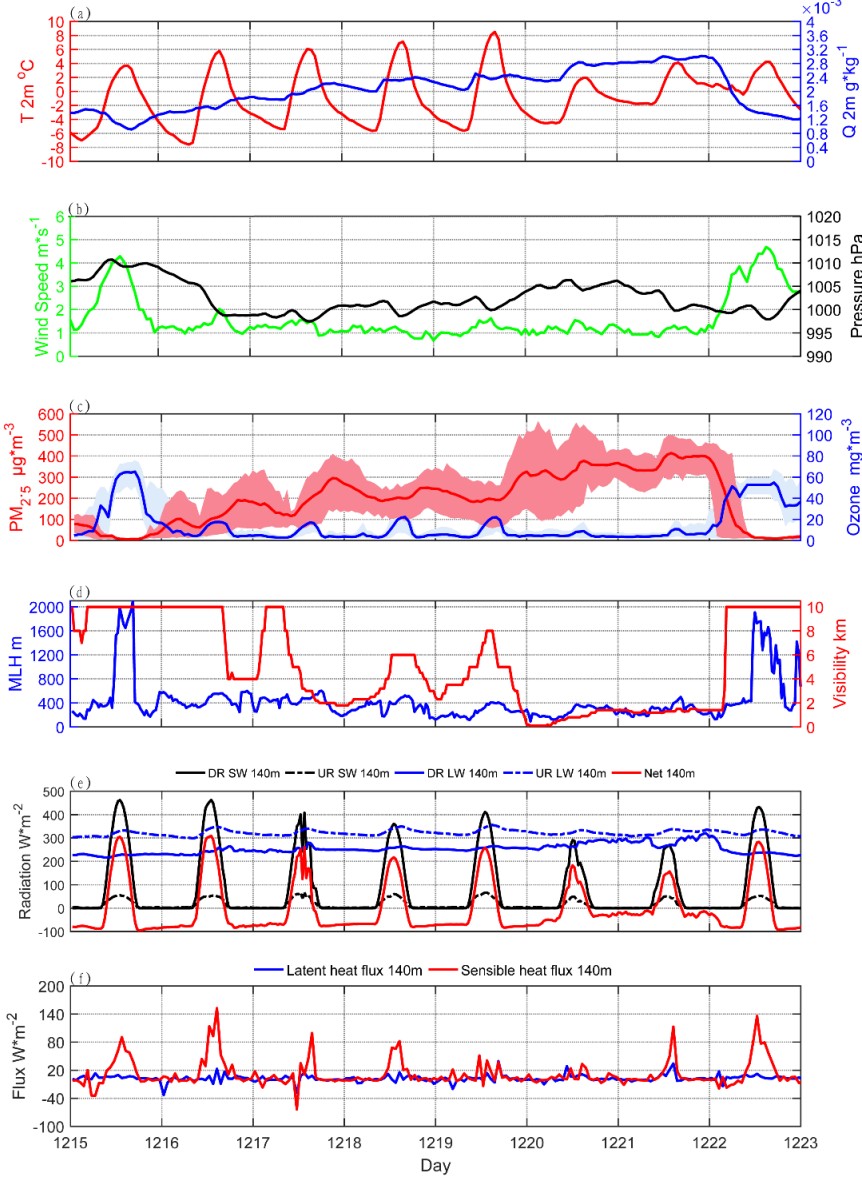

576

Figure 5 Diurnal pattern of observed variables from the 15th to 23rd of December in Beijing. (a) Temperature (red line; °C) and absolute humidity (blue line; g kg$^{-1}$) at 2 m; (b) wind speed at 10 m (green line; m s$^{-1}$) and pressure (black line; hPa); (c) average PM$_{2.5}$ concentration (red line is the average and the shading indicates the standard deviation; ug m$^{-3}$) and ozone concentration (blue lines and the shading indicate the standard deviation; mg m$^{-3}$) of 35 environmental monitoring stations in Beijing; (d) mixing layer height (blue line; m) and visibility (red line; km); (e) radiation from the observation tower at 140 m, downward shortwave radiation (solid black line; W m$^{-2}$), upward shortwave radiation (dashed black line; W m$^{-2}$), downward longwave radiation (solid blue line; W m$^{-2}$), upward longwave radiation (dashed blue line; W m$^{-2}$), net radiation (red line; W m$^{-2}$); and (f) sensible heat flux (red line; W m$^{-2}$) and latent flux (red line; W m$^{-2}$).







Figure 6 Diurnal pattern of the simulated variable from the 15th to 23rd of December. (a) Temperature at 2 m (℃); (b)

specific humidity (g kg⁻¹) at 2 m; (c) shortwave radiation (W m⁻²); (d) longwave radiation (W m⁻²); (e) MLH (m); (f)
sensible heat flux (W m⁻²); and (g) latent heat flux (W m⁻²).






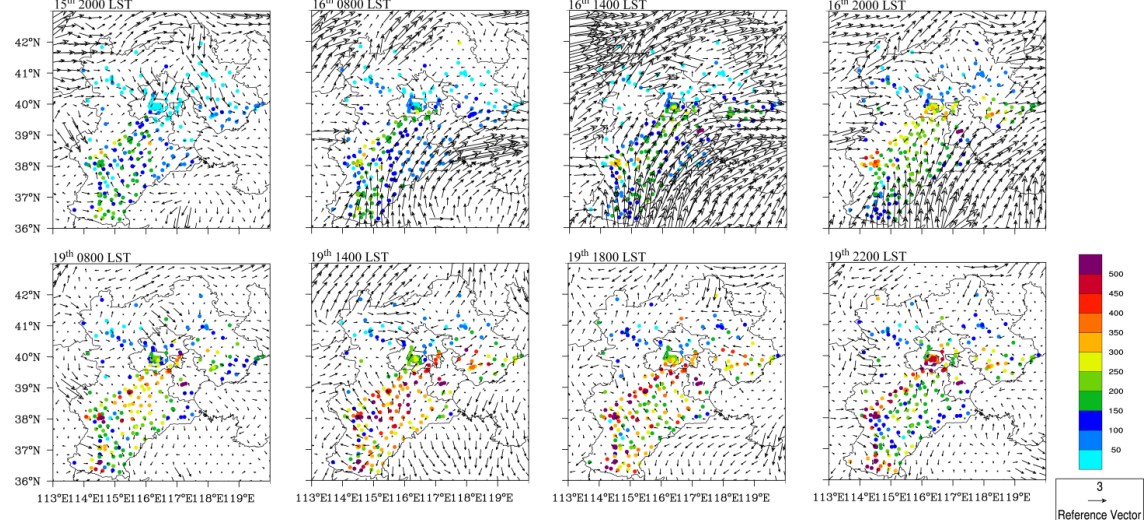


Figure 7 Spatial distribution of the observed concentration of PM$_{2.5}$ (dots; ug m$^{-3}$) and wind field (vector; m s$^{-1}$) for two
increasing processes of the concentration of PM$_{2.5}$.




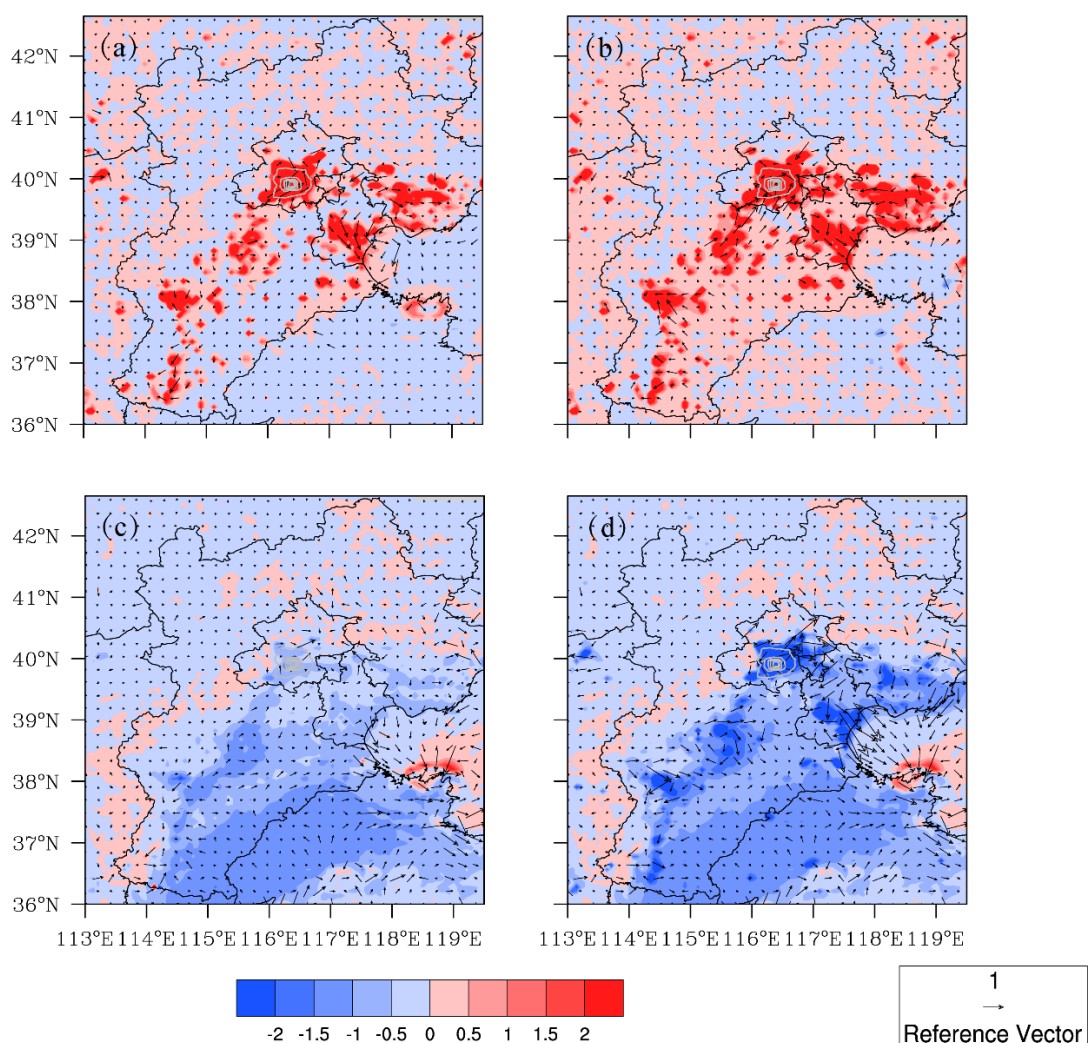


Figure 8 Spatial distribution of simulated temperature (shading; °C) and wind field (vector; m s$^{-1}$). (a) UI_aero; (b) UI_noaero;

(c) AI_urban; (d) AI_nourban.






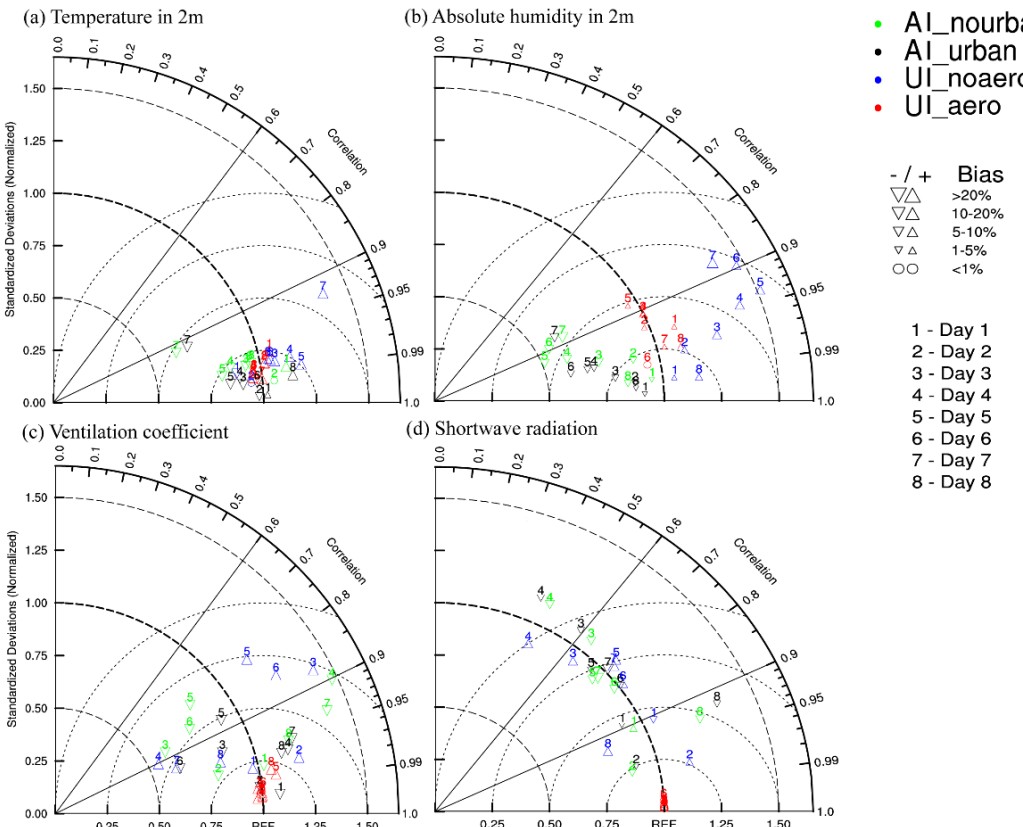


Figure 9 Daily means of the four types of impacts (UI_aero, UI_noaero, AI_urban, AI_nourban) in the eight days are shown
in Taylor diagrams in the Beijing area. (a) Temperature at 2 m (°C); (b) absolute humidity (g kg$^{-1}$); (c) ventilation coefficient
(m$^2$ s$^{-1}$); (d) shortwave radiation (W m$^{-2}$).

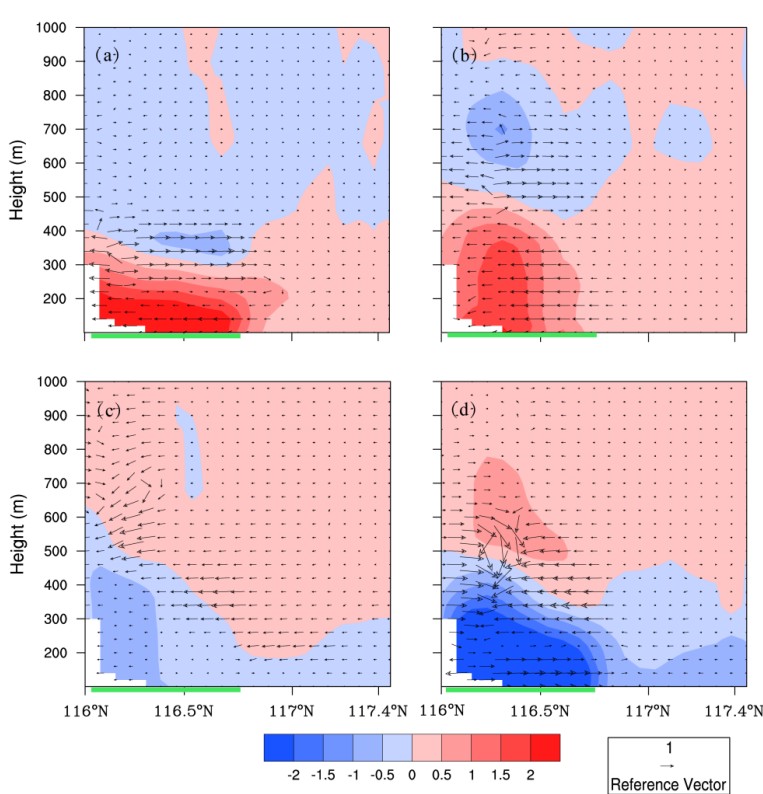

Figure 10 Cross section at 39.9 °N of average temperature (shading; °C) and wind field (vector; m s$^{-1}$) from 0000 LST to 0800 LST on the 16th to 20th. (a) UI_aero; (b) UI_noaero; (c) AI_urban; (d) AI_nourban.