# Peer review of "The interaction between urbanization and aerosols during a haze event"

_Atmospheric Chemistry and Physics, 2020_

## Referee Comment (RC1) · Anonymous Referee #1 · 25 Apr 2020

The interaction between aerosols and urbanization during haze events is investigated due to the surface energy balance using the Rapid-Refresh Multiscale Analysis and Prediction System-Short Term (RMAPS-ST). Aerosols reduce urban-related warming during the daytime. It was found that atmospheric warming decreased by 30 to 50% if the concentration of PM2.5 increased from 200 to 400 $\mu$gÂům-3. Aerosols enhance the urban-related atmospheric warming at night during such an PM2.5 increase by approximately 28%. This is important for haze formation. Urbanization reduced the aerosol-related cooling effect by approximately 54% during the haze event, and the strength of the impact changed little with increasing aerosol content. The impact of aerosols on urban-related warming is more significant than the impact of urbanization on aerosol-related cooling. Aerosols decreased the urban-impact on the mixing layer

height by 148% and on the sensible heat flux by 156%. Furthermore, the aerosols decreased the latent heat flux, and the impact was reduced by 48.8% due to urbanization. The impact of urbanization related changes in on the reduction of pollutant concentrations is more important than that of aerosols. The radiation interaction between urbanization and aerosols may enhance the accumulation of pollution and decrease transport of pollution. General comments A general description of physical processes between aerosols or PM2.5 and warming and cooling are missing in the abstract. A more general discussion of the atmospheric physics which is studied here is required to understand what the authors want to tell us. This topic is much better handled in the chapter Introduction. But the last sentence of the Introduction is producing questions so that this statement should be deleted here but discussed in the chapter Conclusions. The description of methods is missing an overall statement which data are required and why. There it is necessary also to show what is available and which data are missing. It should be explained why the data basis is complete for this study. Then the algorithms and models should be discussed by the same view: why you do what and why this way can provide the expected results or answers to the hypothesis. The description of results is very detailed so that more information for understanding is required as mentioned above. The chapter Conclusions are a summary, a discussion and some conclusions. The discussion is missing the relation of the study results to the overall knowledge. What is new? What are the conclusions for the overall knowledge and the study area? The paper addresses relevant scientific questions within the scope of ACP. The paper presents novel concepts, ideas, tools and data. The scientific methods and assumptions are valid and clearly outlined so that substantial conclusions are reached. The description of experiments and calculations allow their reproduction by fellow scientists. The results are sufficient to support the interpretations and conclusions. The quality of the figures is good. The figure captions should be improved so that these are understandable without the overall manuscript: terms must be explained, description of parameters (Fig. 2c). The related work is well cited so that the authors give proper credit to related work and own new contribution. The title reflects the whole content of

the paper. The abstract must be improved (see above) to provide a concise and complete summary. The overall presentation is well structured and clear. The language is fluent and precise but must be improved in very much details. It is necessary that a native speaker is improving the manuscript. The mathematical formulae, symbols, abbreviations, and units are generally correctly defined and used. No parts of the paper (text, formulae, figures, tables) should be reduced, combined, or eliminated. The number and quality of references is appropriate. Specific Comments Please follow the guidelines to write the references: the authors of papers are incomplete, after the title you set a"." or a ",", some paper references include the doi number and other not. Technical corrections Line 76 Crutzen instead of Cruten.

---

## Referee Comment (RC2) · Anonymous Referee #2 · 22 May 2020

Summary The authors investigate the interaction between aerosols and urbanization during a severe haze event via the RMAPS-ST model. Results indicate that a 100% increase in PM2.5 (200 to 400 $\mu$g/m3) reduced daytime urban-related warming by 20% (from 30-50%). However, urban-related warming increased approximately 28% in response to aerosols- important for haze formation. With regards to urbanization, the aerosol-related cooling effect was reduced by approximately 54%, changing little with aerosol increases. The study also found that aerosols reduced the urban-impact on the mixing layer, sensible heat flux, and latent heat flux by 148%, 156%, and 48.8%, respectively. This reviewer's main concern is related to whether or not the authors address aerosol typology in the model. If so aerosol chemistry was considered, then how? The work could be greatly improved with better section transitions, and by addressing several items described below.

Abstract- Which haze event? The authors should specify. Lines 30-33: Rephrase for better flow. Lines 37-38: Unclear.

Introduction- The authors thoroughly cite references to support statements and do a good job of showing the importance of aerosol-urban impacts. They also state that quantitative evaluation of urban impacts on aerosols and vice-versa has not been conducted simultaneously in metropolitan areas. There are several sentences that need to be rephrased- some of which are listed below. Lines 43-46: Rephrase to improve the flow. Lines 49-54: These lines can be connected better connected. Lines 74-75: Rephrase. Lines 87-88: Which "conclusions" specifically? Line 103: Add the word "model" after (RMAPS-ST) Line 104: Remove "the mechanism of"

Methods- The authors immediately describe four observational data types used for the study and provide a map of the locations (in Figure 1, is the shaded region topography? What units?). This reviewer was expecting a mention of the high RMSE values for longwave and shortwave (Table 1). What is this attributed to? Line 113: Rephrase to "synoptic conditions" Lines 143-154: What considerations were made for other important aerosol parameters such aerosol particle size distribution and typology?

Results- The authors first describe the haze 15-22 December 2016 haze event, thoroughly describing the evolution of the event in three stages. The specifics of the simulation are then described, but this section should be moved to Methodology (Section 3.2). Simulation results are then described. There are so many numbers in the results section that an additional table could be added. The authors could also organize the results better, as it is a bit confusing going back and forth from aerosol impact on the urban to urban impacts on the aerosol. 246 pollution and by up to 56% under heavy pollution Line 167: What makes a heavy haze event typical? Lines 194: "on" the morning of... Lines 222-226: Rephrase, and also consider replacing the word "obviously". Figure 6: Are these results averaged over a specific grid? Lines 270-271: What is

meant by "a few differences"? Lines 308-309: I think I understand what you're saying here, but this needs to be clearer. Line 329: wind fields "are" very important.

Conclusion- The authors summarize their findings and highlight the most important results. The paper ends without the authors discussing the implications of their findings their findings, and could benefit from such a discussion being added. Line 379: Why not just list the actual maximum concentration?

Figures- Figure 3: Is difficult to see, the red dashed contours are not clear on the panels. Figure 4: Add units on the left axis. Also, consider using a box instead of the extra shaded regions on the 16th, 17th, and 19th. Figure 9: This reviewer finds this plot to be well put together (just wanted to say that!).

---

## Author Comment (AC1) · 16 Jun 2020

Dear reviewer:

Thank you for your comments concerning our manuscript entitled "The interaction between urbanization and aerosols during the typical haze event". The comments are all valuable for improving the manuscript and also have great guiding significance for our research. We have studied the comments carefully and made corrections that we hope will be met with your approval. One version of the revised manuscript is highlighted with Track Changes. In the following we quoted each review question and added our response after each paragraph.
* * *
**Reviewer #1:**

*General comments:*

1. *A general description of physical processes between aerosols or PM$_{2.5}$ and warming and cooling are missing in the abstract. A more general discussion of the atmospheric physics which is studied here is required to understand what the authors want to tell us.*

Thank you for your suggestion. We added a general description of warming and cooling processes by aerosols or PM$_{2.5}$ in the Abstract to improve the expression of physical mechanisms in the revised manuscript.

The new part was added in Lines 27-29 in the revised manuscript:

Aerosols cause cooling at the surface by reducing shortwave radiation, while urbanization causes warming by altering the surface albedo and releasing anthropogenic heat. The combined effect of the two phenomena needs to be studied in depth.

2. *This topic is much better handled in the chapter Introduction. But the last sentence of the Introduction is producing questions so that this statement should be deleted here but discussed in the chapter Conclusions.*

We deleted the last sentence of the Introduction and added it to the Discussion section in the revised manuscript.

3. *The description of methods is missing an overall statement which data are required and why. There it is necessary also to show what is available and which*

*data are missing. It should be explained why the data basis is complete for this study. Then the algorithms and models should be discussed by the same view: why you do what and why this way can provide the expected results or answers to the hypothesis. The description of results is very detailed so that more information for understanding is required as mentioned above.*

Thank you for your suggestion. We added more information and reorganized the Methods section to explain the data basis.

The revised Methods section is as follows (the added parts are shown in red):

**2 Methods**

**2.1 Observational data**

To investigate the interaction between urbanization and aerosols, observation data on basic meteorological elements, air quality, radiation and surface heat flux and the mixing layer height (MLH) are very important to reveal the impact of urbanization and aerosols during haze events.

The basic meteorological elements were obtained from 309 national basic weather stations in the BTH region and were provided by the China Meteorological Administration (http://data.cma.cn/). The locations of the national basic weather stations are shown in Fig 1 (red dots). The mass concentrations of fine particulate matter ($PM_{2.5}$) were recorded by 251 environmental monitor stations managed by the Ministry of Ecology and Environment of the People's Republic of China (http://hbk.cei.cn/aspx/default.aspx) (Fig 1, black dots). We also used radiation and surface heat flux data to analyze the urban surface energy budget obtained from the Beijing meteorological tower (39.97°N, 116.37°E). The tower is 325 m high and is operated by the Institute of Atmospheric Physics (IAP), Chinese Academy of Sciences (CAS). The heat flux data were measured by a fast response eddy covariance sensor system that was sampled at 10 Hz using CR500 (Campbell Scientific Inc., USA). The radiation data were provided by Kipp & Zonen (Netherlands) four-component unventilated CNR1 radiometers. Radiation and surface flux data from 140 m of the tower were used in this study. In addition, the MLH is an important factor affecting pollutant diffusion and is also affected by both urbanization and aerosols. Because the

MLH is not a routine observation, we obtained the data from only one site. The MLH and backscattering coefficient were measured by enhanced single-lens ceilometers (Vaisala, CL51, Finland) deployed by the IAP (Tang et al., 2016). Backscattering coefficient profiles were calculated by referencing the attenuation strobe laser LiDAR technique (910 nm), which is cited in Tang et al. (2015).

**2.2 Model description and experimental design**

To investigate the respective effects of urbanization and aerosols and further determine the interaction between urbanization and aerosols, a high-resolution regional model with satisfactory performance is necessary for sensitivity tests.

…

4. *The chapter Conclusions are a summary, a discussion and some conclusions. The discussion is missing the relation of the study results to the overall knowledge. What is new? What are the conclusions for the overall knowledge and the study area?*

Thank you for your suggestion. We added a Discussion section to show the innovations and the relation of the study results to the overall knowledge.

The Discussion section is as follows:

**5 Discussion**

In this study, it was easier to distinguish the impacts of aerosols and urbanization by using RMAPS-ST with AOD hourly inputs than with RMAPS-Chem. One reason for this difference is that the model performance of RMAPS-ST is much better than that of RMAPS-Chem in meteorological fields. Although real-time feedback in modeling is not provided, RMAPS-ST is more efficient and more suitable for short-term operational forecasting.

This study not only qualified the impacts of aerosols and urbanization on haze events but also analyzed the interaction between aerosols and urbanization during haze events. This research will help to improve air quality under the continuous urbanization and sustainable development of large cities.

The government has taken a series of emission reduction measures, including limiting

industrial emissions and vehicle plate number traffic restriction measures, to improve the air quality in the BTH region. The policies have been effective in reducing aerosols. At the same time, urbanization continues mainly in the areas around Beijing (such as the Xiongan New Area). The results of this study show that the combined impact of urbanization and decreasing aerosols will increase the downward shortwave radiation and further increase the surface temperature and ozone concentration in the boundary layer. Previous studies indicated that ozone generally increases with temperature and decreases with humidity (Camalier et al., 2007; Cardelino et al., 1990). It is well known that ozone is not only a pollutant but also a greenhouse gas. Therefore, ozone will form a positive feedback mechanism to induce warming and ozone pollution in the boundary layer. This feedback will pose a new challenge regarding how to reduce ozone pollution in urban areas. Some studies have suggested that urban greening can effectively reduce ozone pollution (Nowak et al., 2000; Benjamin and Winer, 1998). More attempts should be made to add the interaction between urbanization and ozone in regional models.

**Reference**

Camalier, L., Cox, W., and Dolwick, P.: The effects of meteorology on ozone in urban areas and their use in assessing ozone trends, Atmospheric Environment, 41(33), 7127-7137, 2007.

Cardelino, C. A., and Chameides, W. L.: Natural hydrocarbons, urbanization, and urban ozone, Journal of Geophysical Research, 95(D9), 13971, 1990.

Nowak, D. J., Civerolo, K. L., Rao, S. T., Sistla, G., Luley, C. J., and Crane, D. E.: A modeling study of the impact of urban trees on ozone, Atmospheric Environment, 34(10), 1601-1613., 2000.

Benjamin, M. T., Winer, A. M.: Estimating the ozone-forming potential of urban trees and shrubs, Atmospheric Environment, 32(1), 53-68, 1998.

5. *The figure captions should be improved so that these are understandable without the overall manuscript: terms must be explained, description of parameters (Fig. 2c).*

Thank you for your suggestion. The revised Fig 2 is as follows:

[Figure]

Figure 2 (a) Hourly backscattering coefficient (shading; Mm·sr$^{-1}$) observed by single-lens ceilometers (39.97°N, 116.37°E) from the 15$^{th}$ to 23$^{rd}$ of December; (b) hourly column backscatter coefficient (black line; sr$^{-1}$) and AOD used in modeling for Beijing (blue line) and (c) scatter diagram of hourly column backscatter coefficient and AOD (blue dots) and their correlations (red line).

6. *Please follow the guidelines to write the references: the authors of papers are incomplete, after the title you set a".” or a “,”, some paper references include the doi number and other not. Technical corrections Line 76 Crutzen instead of Cruten.*

Thank you for your suggestion. We unified the format and added information to improve the References section.
* * *
Special thanks to you for your good comments. We tried our best to improve the manuscript and made some changes in the manuscript. These changes will not influence the content and framework of the paper. Furthermore, to make the article more readable, we have had the manuscript polished with a professional assistance in writing.

We appreciate for Reviewer' warm work earnestly, and hope that the correction will meet with approval.

Once again, thank you very much for your comments and suggestions.

Yours sincerely,

Dr. Tang

---

## Author Comment (AC2)

Dear reviewer:

Thank you for your comments concerning our manuscript entitled "The interaction between urbanization and aerosols during the typical haze event". The comments are all valuable for improving the manuscript and also have great guiding significance for our research. We have studied the comments carefully and made corrections that we hope will be met with your approval. One version of the revised manuscript is highlighted with Track Changes. In the following we quoted each review question and added our response after each paragraph.
* * *
**Reviewer #2:**

*The authors investigate the interaction between aerosols and urbanization during a severe haze event via the RMAPS-ST model. Results indicate that a 100% increase in PM2.5 (200 to 400 _g/m3) reduced daytime urban-related warming by 20% (from 30-50%). However, urban-related warming increased approximately 28% in response to aerosols- important for haze formation. With regards to urbanization, the aerosol-related cooling effect was reduced by approximately 54%, changing little with aerosol increases. The study also found that aerosols reduced the urban-impact on the mixing layer, sensible heat flux, and latent heat flux by 148%, 156%, and 48.8%, respectively. This reviewer's main concern is related to whether or not the authors address aerosol typology in the model. If so aerosol chemistry was considered, then how?*

Thank you for your suggestion. The aerosol typology has been considered in this study. The AOD was extracted from the output of RMAPS-Chem (Zhao et al., 2019; Zhang et al., 2018), which included the aerosol typology in the model. Then, we added the hourly distribution of AOD in the RRTMG radiation scheme in RMAPS-Urban. The particle size distribution and typology of aerosols also calculated in the RRTMG radiation scheme is according to Ruiz et al. (2014). Therefore, the particle size distribution and typology of aerosols are included in both the input hourly AOD fields and the RRTMG radiation scheme.

We added the sentence "The particle size distribution and typology of aerosols used in

this study is according to Ruiz et al. (2014)" in Lines 153-154 to clarify this
information.

*The work could be greatly improved with better section transitions, and by addressing
several items described below.*

*1.  Abstract:*

*a)  Which haze event? The authors should specify.*

We added information on the haze event in Line 30.

Line 30: The interaction between aerosols and urbanization during the haze event that
occurred from the 15th to 22nd of December 2016 in Beijing was investigated using the
rapid-refresh multiscale analysis and prediction system-short term (RMAPS-ST).

*b)  Lines 30-33: Rephrase for better flow.*

Aerosols reduced urban-related warming during the daytime. The urban-related
warming decreased by 30 to 50% as the concentration of PM2.5 increased from 200
to 400 $\mu g \cdot m^{-3}$. Conversely, aerosols also enhanced urban-related warming at dawn,
and the increment was approximately 28%, which contributed to haze formation.

*c)  Lines 37-38: Unclear.*

Furthermore, aerosols decreased the latent heat flux; however, this reduction decreased
by 48.8% due to urbanization.

*2.  Introduction-The authors thoroughly cite references to support statements and do
a good job of showing the importance of aerosol-urban impacts. They also state
that quantitative evaluation of urban impacts on aerosols and vice-versa has not
been conducted simultaneously in metropolitan areas. There are several sentences
that need to be rephrased- some of which are listed below.*

We revised the Introduction section according to your suggestions.

*a)  Lines 43-46: Rephrase to improve the flow.*

In recent years, heavy haze pollution events have increasingly occurred in densely
populated urban areas, such as the Beijing-Tianjin-Hebei region (BTH region) and
Yangtze River Delta region of China (Zhang et al., 2019). These events have caused
increasingly severe adverse effects on transportation, the ecological environment and

human health (Zhao et al., 2012; Wu et al., 2010; Liu et al., 2012).

*b)    Lines 49-54: These lines can be connected better connected.*

The revised version: The conditions for the formation of heavy haze in the BTH region are very complex (Miao et al., 2017; Wei et al., 2018; Ren et al., 2019). Although emissions, meteorological conditions, terrain, and high-density human activities in urban areas are all important conditions for the evolution of heavy haze (Huang et al., 2008a; Zhu et al., 2018), meteorological conditions are critical for the evolution of heavy haze pollution weather under the background of constant emissions (Wang et al., 2020; Pei et al., 2020).

*c)    Lines 74-75: Rephrase.*

The revised version: However, in contrast to the effects of urbanization, aerosols cause cooling at the surface by reducing shortwave radiation to enhance static stability (Grimmond, 2007; Cruten, 2004, Huang et al., 2007).

*d)    Lines 87-88: Which "conclusions" specifically?*

Xu et al. (2019) indicated that the impact of irrigation on regional climate may vary depending on the scale. We cited Xu et al. (2019) to explain that the different conclusions obtained by Cao et al. (2016) and Yang et al. (2020) may be due to the focus on different scales.

*e)    Line 103: Add the word "model" after (RMAPS-ST)*

The suggested change has been made.

*f)   Line 104: Remove "the mechanism of"*

The suggested change has been made.

*3.  Methods:*

*a)   The authors immediately describe four observational data types used for the study and provide a map of the locations (in Figure 1, is the shaded region topography? What units?).*

We improved the caption of Figure 1 to clarify this information.

The revised capture: Figure 1 Domain configuration of RMAPS-ST and the location of the study area, indicated by the solid white line. The black dots indicate the locations

of the 251 environmental monitoring stations, and the red dots represent the 309 meteorological stations in the BTH region, where the gray loop lines show the locations of the second to sixth ring roads. The shading is the terrain height (unit: m).

b) *This reviewer was expecting a mention of the high RMSE values for longwave and shortwave (Table 1). What is this attributed to?*

There are two possible reasons for the high RMSE values for longwave and shortwave radiation:

i) Deficiency of observation sites and interpolation methods

Only observed longwave and shortwave data from the Beijing meteorological tower (39.97°N, 116.37°E) were available for evaluation. The weighted interpolation of the nine points was used to transfer the grid modeling results to the station locations. A total of 294 observation stations were used to evaluate basic meteorological elements such as temperature. The RMSE of the basic meteorological elements is the average of the 294 observation stations. Therefore, it is reasonable that the RMSE values of the radiation and heat flux values are larger than those of basic meteorological elements.

The magnitudes of longwave and shortwave radiation are larger than that of heat flux (Fig 5e and f). Although the RMSE of radiation is larger than that of heat flux, the absolute error ratio is similar.

ii) Height differences between observations and simulations

Observed shortwave and longwave radiation data from the tower were only available from 140 m. However, the surface radiation was simulated from the shortwave and longwave radiation.

We added an explanation in the revised version as follows.

Lines 171-173: The deficiency of observation sites, interpolation methods and the height differences between the observations and simulations resulted in higher root mean square error (RMSE) values for radiation and heat flux than for the other variables.

c) *Line 113: Rephrase to "synoptic conditions"*

We deleted this sentence in the revised manuscript.

*d) Lines 143-154: What considerations were made for other important aerosol parameters such aerosol particle size distribution and typology?*

Aerosol particle size distribution and typology:

Ruiz et al. (2014) elaborated on how to specify the AOD at each spectral band in the RRTMG scheme. A 2-band version of the Ångström law (Gueymard, 2001) was used as follows:

$$\tau(\lambda) = \tau 0.55(\frac{\lambda}{0.55})^{-\alpha_i}$$

where $\lambda$ is the wavelength in μm and $\alpha_i$ is the Ångström exponent for each band, defined as $\alpha_i = \alpha_1$ for _<0.55 μm, and $\alpha_i=\alpha_2$ otherwise. The corresponding values of $\alpha_i$ are given in Table 2. For $\alpha_1$, extinction coefficients of 0.337, 0.55 and 0.649 μm were used. The values at 0.55, 0.649, 1.06 and 1.536 μm were used for $\alpha_2$.

We added an explanation of the aerosol particle size distribution and typology in the new version as follows.

Lines 153-154: The particle size distribution and typology of aerosols used in this study is according to Ruiz et al. (2014).

**Reference**

Ruiz-Arias, J. A., Dudhia, J., and Gueymard, C. A. (2014). A simple parameterization of the short-wave aerosol optical properties for surface direct and diffuse irradiances assessment in a numerical weather model. Geoentific Model Development, 7(3), 1159-1174.

*4. Results:*

*a) The authors first describe the haze 15-22 December 2016 haze event, thoroughly describing the evolution of the event in three stages. The specifics of the simulation are then described, but this section should be moved to Methodology (Section 3.2).*

Thank you for your suggestion. We first showed the weather maps and time series of meteorological elements in Section 3.1 from observations, namely, what the observations told us. However, we begin to design sensitivity tests and analyze the modeling results in Section 3.2. Therefore, we changed the chapter title to "3.1 Observation and weather condition analysis" to make it clear.

*b) Simulation results are then described. There are so many numbers in the results*

*section that an additional table could be added.*

We added Table 3 to summarize the numbers.

Table 3 Quantitative results of the interaction between urbanization and aerosols

| Time | Temperature °C | | Specific humidity ×10-2 g kg⁻¹ | | Longwave W·m⁻² | | MLH m | Sensible heat flux W·m⁻² | Latent heat flux W·m⁻² |
|---|---|---|---|---|---|---|---|---|---|
| | 16th-19th | 20th-21st | 16th-19th | 20th-21st | 16th-19th | 20th-21st | 16th-21st | 16th-21st | 16th-21st |
| UI_aero | 0.42 | 0.19 | 3.66 | 3.08 | 0.10 | -0.02 | -1.97 | -1.01 | 0.03 |
| UI_noaero | 0.60 | 0.35 | 4.78 | 4.48 | 0.62 | 0.51 | 4.04 | 1.74 | 0.49 |
| AI_urban | -0.16 | -0.19 | -0.88 | | -0.24 | | -4.37 | -1.64 | -0.50 |
| AI_nourba | -0.34 | -0.43 | 1.36 | | -0.73 | | -10.38 | -4.02 | -0.96 |

c) *The authors could also organize the results better, as it is a bit confusing going back and forth from aerosol impact on the urban to urban impacts on the aerosol.*

Thank you for your suggestion. We unified the order of the analysis to show the impacts of aerosols on urban areas first for each variable and added Table 3 to clarify this information in the revised manuscript.

d) *Line 167: What makes a heavy haze event typical?*

Large-scale weather conditions result in poor dispersion of pollutants are the main factor of typical continuous severe heavy haze formation.

e) *Lines 194: "on" the morning of: : :*

The suggested change has been made

f) *Lines 222-226: Rephrase, and also consider replacing the word "obviously".*

The revised version: The impact of urbanization on the near-surface temperature displays diurnal variation in the Beijing area. The warming effect of urbanization was dominant at night. The urban impact on temperature was partly offset under aerosol conditions when comparing the results of UI_aero and UI_noaero, especially during the daytime (Fig 6a, red lines).

g) *Figure 6: Are these results averaged over a specific grid?*

The results are processed to the regional average for the Beijing area.

h) *Lines 270-271: What is meant by "a few differences"?*

"a few differences" means the difference was very small. We revised the sentence to "Aerosols reduce the downward shortwave radiation during the daytime, and the

differences between AI_urban and AI_nourban are very small." to clarify this information.

i) *Lines 308-309: I think I understand what you're saying here, but this needs to be clearer.*

We revised the sentence to the following: The above results indicate that the offsetting effect of aerosols on urbanization is more important than the impact of urbanization on aerosols on local weather.

j) *Line 329: wind fields "are" very important.*

The suggested change has been made.

5. *Conclusion*

a) *The authors summarize their findings and highlight the most important results. The paper ends without the authors discussing the implications of their findings their findings, and could benefit from such a discussion being added.*

We added a Discussion section in the new version as follows.

**5 Discussion**

In this study, it was easier to distinguish the impacts of aerosols and urbanization by using RMAPS-ST with AOD hourly inputs than with RMAPS-Chem. One reason for this difference is that the model performance of RMAPS-ST is much better than that of RMAPS-Chem in meteorological fields. Although real-time feedback in modeling is not provided, RMAPS-ST is more efficient and more suitable for short-term operational forecasting.

This study not only qualified the impacts of aerosols and urbanization on haze events but also analyzed the interaction between aerosols and urbanization during haze events. This research will help to improve air quality under the continuous urbanization and sustainable development of large cities.

The government has taken a series of emission reduction measures, including limiting industrial emissions and vehicle plate number traffic restriction measures, to improve

the air quality in the BTH region. The policies have been effective in reducing aerosols. At the same time, urbanization continues mainly in the areas around Beijing (such as the Xiongan New Area). The results of this study show that the combined impact of urbanization and decreasing aerosols will increase the downward shortwave radiation and further increase the surface temperature and ozone concentration in the boundary layer. Previous studies indicated that ozone generally increases with temperature and decreases with humidity (Camalier et al., 2007; Cardelino et al., 1990). It is well known that ozone is not only a pollutant but also a greenhouse gas. Therefore, ozone will form a positive feedback mechanism to induce warming and ozone pollution in the boundary layer. This feedback will pose a new challenge regarding how to reduce ozone pollution in urban areas. Some studies have suggested that urban greening can effectively reduce ozone pollution (Nowak et al., 2000; Benjamin and Winer, 1998). More attempts should be made to add the interaction between urbanization and ozone in regional models.

**Reference**

Camalier, L., Cox, W., and Dolwick, P.: The effects of meteorology on ozone in urban areas and their use in assessing ozone trends, Atmospheric Environment, 41(33), 7127-7137, 2007.

Cardelino, C. A., and Chameides, W. L.: Natural hydrocarbons, urbanization, and urban ozone, Journal of Geophysical Research, 95(D9), 13971, 1990.

Nowak, D. J., Civerolo, K. L., Rao, S. T., Sistla, G., Luley, C. J., and Crane, D. E.: A modeling study of the impact of urban trees on ozone, Atmospheric Environment, 34(10), 1601-1613., 2000.

Benjamin, M. T., Winer, A. M.: Estimating the ozone-forming potential of urban trees and shrubs, Atmospheric Environment, 32(1), 53-68, 1998.

*b) Line 379: Why not just list the actual maximum concentration?*

Line 379 to Line 403: We rephrased this sentence and added the actual maximum concentration of $PM_{2.5}$. The revised sentence: The average concentration of $PM_{2.5}$ was approximately 200 $\mu g \cdot m^{-3}$, and the maximum was 695 $\mu g \cdot m^{-3}$.

*6. Figures:*

a) *Figure 3: Is difficult to see, the red dashed contours are not clear on the panels.* We improved the quality of Figure 3 to make it clear.

[Figure]

Figure 3 Weather maps. (a) 0800 LST on the 16th at 700 hPa; (b) 0800 LST on the 18th at 700 hPa; (c) 0800 LST on the 19th at 700 hPa; (d) 2000 LST on the 19th at 700 hPa; (e) 0800 LST on the 16th at 850 hPa; (f) 800 LST on the 18th at 850 hPa; (g) 0800 LST on the 19th at 850 hPa; (h) 2000 LST on the 19th at 850 hPa.

b) *Figure 4: Add units on the left axis. Also, consider using a box instead of the extra shaded regions on the 16th, 17th, and 19th.*

We added the units and replaced the shading with a box in Figure 4.

[Figure]

Figure 4 Hourly wind profile from the 15th to 23rd of December. Wind speed (shading; m·s$^{-1}$) and horizontal wind field (vector; m·s$^{-1}$). The black boxes show the two periods of south wind conveyance.
* * *
Special thanks to you for your good comments. We tried our best to improve the manuscript and made some changes in the manuscript. These changes will not influence the content and framework of the paper. Furthermore, to make the article more readable,

we have had the manuscript polished with a professional assistance in writing.

We appreciate for Reviewer' warm work earnestly, and hope that the correction will meet with approval.

Once again, thank you very much for your comments and suggestions.

Yours sincerely,

Dr. Tang

---

## Referee Report (RR1)

**The interaction between urbanization and aerosols during the haze event.**

**Miao Yu1 , Guiqian Tang2 , Yang Yang1 , Shiguang Miao1 , Yizhou Zhang1 , Qingchun Li1**

The authors investigate the interaction between aerosols and urbanization during a severe haze event via the RMAPS-ST model. Results indicate that a 100% increase in $PM_{2.5}$ (200 to 400 µg·m$^{-3}$) reduced daytime urban-related warming by 20% (from 30-50%). However, urban-related warming increased approximately 28% in response to aerosols- important for haze formation. With regards to urbanization, the aerosol-related cooling effect was reduced by approximately 54%, changing little with aerosol increases. The study also found that aerosols reduced the urban-impact on the mixing layer, sensible heat flux, and latent heat flux by 148%, 156%, and 48.8%, respectively. In their revision, the authors appropriately addressed the reviewer suggestions and created an improved manuscript. This reviewer suggests minor changes as follows:

1) Change the title to: "The interaction between urbanization and aerosols during a typical winter haze event in Beijing."
2) Rephrase Lines 29-32: "The effects of urbanization and aerosols were investigated during a typical winter haze event. The event, which occurred in Beijing from 15-22 December 2016, was studied via the rapid-refresh multiscale analysis and prediction system-short term (RMAPS-ST) model."
3) Rephrase from Line 34: "Aerosols reduced urban-related warming during the daytime by 20% (from 30 to 50%) as $PM_{2.5}$ concentrations increased from 200 – 400 µg m$^{-3}$."
4) Rephrase from Line 99: "Cao et al. (2016) describes the first attempt to determine…"
5) Rephrase Lines to 269-271 and add to the previous paragraph.
6) Line 356: replace "smaller" with "less".
7) Paragraph starting from Line 430: Break into two paragraphs, perhaps from line 441.

---

## Author Response (AR2)

Dear editors and reviewer:

Thank you for your comments concerning our manuscript entitled "The interaction between urbanization and aerosols during the typical haze event". The comments are all valuable for improving the manuscript and also have great guiding significance for our research. We have studied the comments carefully and made corrections that we hope will be met with your approval. One version of the revised manuscript is highlighted with Track Changes. In the following we quoted each review question and added our response after each paragraph.

**Reviewer #2:**

The authors investigate the interaction between aerosols and urbanization during a severe haze event via the RMAPS-ST model. Results indicate that a 100% increase in PM2.5 (200 to 400 μg·m-3) reduced daytime urban-related warming by 20% (from 30-50%). However, urban-related warming increased approximately 28% in response to aerosols- important for haze formation. With regards to urbanization, the aerosol-related cooling effect was reduced by approximately 54%, changing little with aerosol increases. The study also found that aerosols reduced the urban-impact on the mixing layer, sensible heat flux, and latent heat flux by 148%, 156%, and 48.8%, respectively. In their revision, the authors appropriately addressed the reviewer suggestions and created an improved manuscript. This reviewer suggests minor changes as follows:

1) *Change the title to: "The interaction between urbanization and aerosols during a typical winter haze event in Beijing."*

The suggested change has been made.

2) *Rephrase Lines 29-32: "The effects of urbanization and aerosols were investigated during a typical winter haze event. The event, which occurred in Beijing from 15-22 December 2016, was studied via the rapid-refresh multiscale analysis and prediction system-short term (RMAPS-ST) model."*

The suggested change has been made.

3) *Rephrase from Line 34: "Aerosols reduced urban-related warming during the daytime by 20% (from 30 to 50%) as PM2.5 concentrations increased from 200 to 400 µg·m$^{-3}$."*

The suggested change has been made.

4) *Rephrase from Line 99: "Cao et al. (2016) describes the first attempt to determine…"*

The suggested change has been made.

5) *Rephrase Lines to 269-271 and add to the previous paragraph.*

The revised the sentence in Lines 276-269: It was not until the strong cold air moved southward in the early morning of the 22$^{nd}$ when the whole atmosphere converted to the northwest stream. The air pollutants were completely removed in the third stage.

6) *Line 356: replace "smaller" with "less".*

The suggested change has been made.

7) *Paragraph starting from Line 430: Break into two paragraphs, perhaps from line 441.*

The suggested change has been made.

[revised manuscript text omitted]

Figure 3 Weather maps. (a) 0800 LST on the 16[th] at 700 hPa; (b) 0800 LST on the 18[th] at 700 hPa; (c) 0800 LST on the 19[th] at 700 hPa; (d) 2000 LST on the 19[th] at 700 hPa; (e) 0800 LST on the 16[th] at 850 hPa; (f) 800 LST on the 18[th] at 850 hPa; (g) 0800 LST on the 19[th] at 850 hPa; (h) 2000 LST on the 19[th] at 850 hPa.

[Figure]

Figure 4 Hourly wind profile from the 15[th] to 23[rd] of December. Wind speed (shading; m·s⁻¹) and horizontal wind field (vector; m·s⁻¹). The black boxes show the two periods of south wind conveyance.

[Figure]

Figure 5 Diurnal pattern of observed variables from the 15th to 23rd of December in Beijing. (a)

Temperature (red line; ℃) and absolute humidity (blue line; g kg⁻¹) at 2 m; (b) wind speed at 10

m (green line; m s⁻¹) and pressure (black line; hPa); (c) average PM₂.₅ concentration (red line is the average and the shading indicates the standard deviation; ug m⁻³) and ozone concentration (blue lines and the shading indicate the standard deviation; mg m⁻³) of 35 environmental monitoring stations in Beijing; (d) mixing layer height (blue line; m) and visibility (red line; km); (e) radiation from the observation tower at 140 m, downward shortwave radiation (solid black line; W m⁻²), upward shortwave radiation (dashed black line; W m⁻²), downward longwave radiation (solid blue line; W m⁻²), upward longwave radiation (dashed blue line; W m⁻²), net radiation (red line; W m⁻²); and (f) sensible heat flux (red line; W m⁻²) and latent heat flux (red line; W m⁻²).

[Figure]

Figure 6 Diurnal patterns of simulated variables from the 15th to 23rd of December. (a)

Temperature at 2 m (℃); (b) specific humidity (g kg$^{-1}$) at 2 m; (c) shortwave radiation (W m$^{-2}$); (d) longwave radiation (W m$^{-2}$); (e) MLH (m); (f) sensible heat flux (W m$^{-2}$); and (g) latent heat flux (W m$^{-2}$).

[Figure]

Figure 7 Spatial distribution of the observed concentration of PM$_{2.5}$ (dots; ug m$^{-3}$) and wind field (vector; m s$^{-1}$) for two increasing processes of the concentration of PM$_{2.5}$.

[Figure]

Figure 8 Spatial distribution of simulated temperature (shading; °C) and wind field (vector; m s$^{-1}$).

(a) UI_aero; (b) UI_noaero; (c) AI_urban; (d) AI_nourban.

[Figure]

Figure 9 Daily means of the four types of impacts (UI_aero, UI_noaero, AI_urban, AI_nourban) in
the eight days are shown in Taylor diagrams in the Beijing area. (a) Temperature at 2 m (°C); (b)
absolute humidity (g kg$^{-1}$); (c) ventilation coefficient (m$^2$ s$^{-1}$); (d) shortwave radiation (W m$^{-2}$).

[Figure]

Figure 10 Cross section at 39.9 °N of average temperature (shading; °C) and wind field (vector; m s⁻

¹) from 0000 LST to 0800 LST on the 16th to 20th. (a) UI_aero; (b) UI_noaero; (c) AI_urban; (d)

AI_nourban.